

# Meteorological analysis of flash floods in Artvin (NE Turkey) on August 24, 2015

Hakki Baltaci[1]

[1] Turkish State Meteorological Service, Istanbul, Turkey

*Correspondence to*:.Hakki Baltaci (baltacihakki@gmail.com)

**Abstract**. On August 24, 2015 intense rainfall episodes generated flash floods and landslides on the eastern Black Sea coast of Turkey. As a consequence of the heavy

rainstorm activity over Artvin and its surroundings (NE Turkey), 11 people died and economic losses totaled a million dollars. During the six hours of the event (from 05:00 UTC to 11:00 UTC), total accumulated rainfall of 136, 64, and 109 mm was measured in the Hopa, Arhavi, and Borçka settlements of Artvin city, respectively. This study comprehensively investigates the meteorological characteristics of those flash floods. In

terms of synoptic mechanisms, the cut-off surface low from the summer Asian monsoon settled over the eastern Black Sea. After two days of quasi-stationary conditions of this cyclone, sea surface temperatures (SSTs) reached 27.5 °C (1.5 °C higher than normal) and low-level moisture convergence developed. In addition, transfer of moisture by warm northerly flows from the Black Sea and relatively cool southerly flows from the land coasts

of the Artvin district exacerbated the unstable conditions, and thus, played a significant role in the development of deep convective cells. Severe rainstorms as well as the slope instability of the region triggered landslides and worsened flood damage in the Artvin.





## 1 Introduction

The interaction between mesoscale convective systems (MCS) on warm Mediterranean Sea and sudden orographic lifting in the coastal regions produces severe precipitation in the Mediterranean countries (Rebora et al., 2012). These severe precipitation events generally generate flash floods and cause serious damages and economic losses. For example, only one single flash flood in the 2002 caused €1.2 billion Euro damages in the Gard region of France (Huet et al., 2003), €300 million Euro damages in Pinios (Greece) flash flood during 1994 (Gaume et al., 2008), €65 million Euro economic losses in Magorala (Spain) flash flood in the 2000 (Llasat et al., 2001), and €4.6 million Euro in the 2007 Mastroguglielmo (Italy) flash flood event (Aronica et al., 2008). Due to its huge social and economic impacts, it is necessary to increase our knowledge about the spatio-temporal dynamics of flash floods to improve their forecast and the land-use planning. For this reason, several studies have analyzed the meteorological (e.g. Milelli et al., 2006; Fragoso et al., 2012), hydrological (e.g. Silvestro et al., 2012) or hydrometeorological (e.g. Delrieu et al., 2005; Borga et al., 2007) characteristics of floods at a particular area and time.

In Turkey, two types of floods generally occur, depending on the catchments characteristics. In the first type, as a consequence of flooding in river basins, large areas are affected and economic losses are considerable (e.g. the overflow of the Meriç river in NW Turkey). The second type, which is more common, is when flash floods are suddenly triggered by severe rainstorms in certain areas (e.g. coastal regions of the country). In this context, numerous studies have investigated the meteorological role in the occurrence of flash floods in different parts of Turkey. Kömüşçü et al. (1998) analyzed the meteorological and terrain features of the flash flood that occurred on November 3 and 4, 1995 on the





Aegean coast, when 61 people died in İzmir (western Turkey). They emphasized that low-level advection, positive vorticity, and strong upper-level divergence together with a squall line oriented NE-SW over the Aegean Sea exacerbated the storm. Subsequently, Kotroni et al., (2006) investigated the storm activity that occurred on December 5, 2002 in Antalya

(southwestern Mediterranean Sea coast of Turkey). They found that warm and moist air masses driven by a low-level jet as well as orographic barriers caused more than 230 mm of 24-h accumulated precipitation during the event. Later, Kömüşcü and Çelik (2013) investigated the hydrometeorological role of a flood at Marmara from September 7 to 10, 2009. They concluded that cold air in the upper atmosphere, a slow-moving quasi-

stationary trough and continuous moisture transfer from the warm Aegean Sea to the surface low were the main mechanisms that led to intense storms.

Differently from the previous studies mentioned above, many severe precipitation events frequently occur and generally conclude with flash floods and trigger landslides in the eastern Black Sea (EBS) region of Turkey (Fig. 1). The EBS comprises the Black Sea (BS)

in the north and the eastern Anatolian Peninsula in the south. The strata of the EBS are generally made of semi-permeable volcanic rocks, which prevent the rainfall from percolation and force the water to flow as runoff (Üçüncü et al., 1994). Located in the north-eastern coast of Turkey, this unique area has the highest mean annual precipitation records (above 2200 mm). The large mountainous area, which is parallel with the BS, and

slope instability as well as intense rainfall result in flash floods and landslide events and threaten the settlements of the region. In addition to all these topographical and meteorological factors, commercial development and urbanization of the region (e.g. the cultivation of tea on the sloping terrain instead of deep-rooted trees and illegal land-usage)





facilitates the flooding. Yüksek et al., (2013) have emphasized that 258 deaths and US $500,000,000 economic losses occurred as a result of the 51 big floods in this basin from 1955 to 2005. They briefly analyzed the hydro-meteorological role of selected nine floods in the region. In one of the latest rainstorm events in the EBS, more than 135 mm of 24-h accumulated rainfall in the Artvin surroundings (i.e. 144, 136 and 149 mm in Hopa, Arhavi and Borçka stations, respectively) caused flash floods and landslides on August 24, 2015, resulting in 11 deaths and a million dollars' worth of economic losses (Fig. 2). In spite of the several negative impacts of flooding for the region and country, there are no detailed studies in the literature which investigate the detailed meteorological role in the development of the convective cells for the EBS. Therefore, the aim of this research is focused on this extreme event, with the following main objectives: (a) to provide a detailed spatio-temporal evaluation of rainstorms on 24 August 2015 that triggered the flash floods and landslides. Daily and hourly precipitation measurements of the available meteorological stations were used to understand temporal and spatial behavior of the rainstorm in the different geographic elevations, (b) to improve our understanding of the meteorological features of this extreme event by focusing on the relevant atmospheric synoptic conditions, satellite and radar images and physical mechanisms (e.g. sea surface temperature evolution) that favored its development.

## 2 Data and Methodology

In order to evaluate the research results, precipitation, sea surface temperature, synoptic and atmospheric data are included in the study. To compare precipitation observations with forecasts, three numerical weather prediction (NWP) model outputs were assessed.





## 2.1 Precipitation and sea surface temperature (SST) data

The eastern Black Sea region is well covered by automated meteorology stations. In addition to the eight long-term stations in the region, 41 new automated meteorology stations have been added since 2013. To present the high spatial resolution as well as

5    retrieve a homogeneous dataset, hourly and daily precipitation data of 49 stations operated by Turkish State Meteorological Service (TSMS) were used in the study (Fig.1). The main characteristics of the stations are described in Table 1.

## 2.2 Synoptic and atmospheric data

The synoptic context of the extreme event of August 24, 2015 as well as the previous day's

10    atmospheric conditions was analyzed with NCEP/NCAR 2.5°X2.5° latitude/longitude reanalysis data. To track the intense rainfall episodes, radar PPI (Plan Position Indicator) images, which provided by TSMS, were used. Rainstorm development stages associated with the flash flood were evaluated with Meteosat 10 images.

## 2.3 Numerical weather prediction (NWP) model outputs

15    Operationally, one global and two regional NWP models are run regularly twice a day (00:00 and 12:00 UTC) for the precipitation forecast by TSMS. In terms of the global NWP, the horizontal grid resolution of ECMWF (European Centre for Medium-Range Weather Forecasts) the IFS (Integrated Forecast System) covers almost 16 km and uses 91 vertical levels. In Alaro, whereas the outer domain has grid spacing of 10 km, the inner

20    domain has almost 5 km of grid spacing as well as 60 vertical levels. The mesoscale NWP



system of WRF (Weather Research and Forecasting) has a horizontal grid spacing of 30 km in its outer computational domain and the inner domain has a grid spacing of 10 km together with 46 vertical levels. To compare precipitation forecasts of these models with the observation results, daily precipitation forecasts of the models belonging on the last runtime

5   for August 24, 2015, at 00:00 UTC outputs were assessed.

## 3 Results and discussion

### 3.1 Precipitation climate of eastern Black Sea

The coastal part of the region is restricted by the EBS Mountain chain in the south and the BS in the north (Fig.1). This mountain chain is in parallel with the BS and has an average

10   altitude of 2000 m. It rises to 3973 m at its highest point (Eris et al., 2012). Apart from the basic synoptic scale circulations such as continental polar and tropical air masses, the region is also affected by orographic precipitation. Colder air masses are prevented by the Caucasus Mountains (the highest point of Georgia) from the north; therefore, moderate climates are seen in the south part of the region. Lee side effects of the mountainous area

15   generate a more continental climate in the southern parts of the EBS (Biyik et al., 2010). In terms of the precipitation values, because of the interactions of weather systems and the orographic uplifting, the highest amount of precipitation is observed during wet and dry seasons compared with the other regions of the country (Unal et al., 2012). To better visualize the seasonal precipitation variability in the EBS, long-term precipitation data from

20   1960 to 2014 were extracted from the available eight meteorology stations (stations marked by stars in Table 1 were used for the climatological approach in Fig. 3). Five stations are



located in the north of the region. According to the results, mean annual precipitation (MAP) changes from 438 mm in the south (Bayburt) to 2243 mm in the north (Hopa). This high spatial precipitation variability generates different land cover terrain. Interestingly, the highest seasonal precipitation amounts were observed in the fall (SON) instead of the

5 winter (DJF) months in the coastal areas. This can be explained by the significance of MCS, flow directions and SST variations over EBS. In the second wettest season (DJF), highest precipitation records were shown for the Hopa, Rize, and Pazar stations with the values 606, 636, and 550 mm, respectively. Nevertheless, the third highest seasonal precipitation totals in these stations are shown in JJA.

**3.2 Spatio-temporal variability of rainfall episodes**

In Fig. 4a, spatial distribution of daily precipitation totals for August 24, 2015 was extracted from 49 meteorological stations. It can be seen that three main cores of precipitation are measured at the Arhavi, Hopa and Borçka stations with the values of 135, 144, and 149 mm, respectively. In Hopa, 27% of the long-term mean of summer rainfalls

was recorded on this day. As a consequence of the intense daily rainfall episodes, these three surrounding areas of Artvin district were those most influenced by flash floods and landslides (i.e. Hopa, Arhavi and Borçka). Among these stations, Hopa (33m altitude, no. 1 in Fig. 1b) is at the lowest altitude and is located in the north coastal part of Artvin city. Borçka station is shown with an altitude of 190m (the second lowest altitude in Artvin, no.

7 in Fig. 1b). The other seaside station, Arhavi (290m altitude, no. 6 in Fig 1b), is located in the northwest and has the third lowest altitude among all Artvin stations. Temporal distribution of these selected stations was extracted as shown in Fig. 4b. Rainstorms started



in the evening (22:00 UTC) of August 23, 2015 and ended at midday on the following day. Hourly observations in the three stations showed the torrential rains increased to 27 to 32 mm between 22:00 and 24:00 UTC on August 23, thereafter suddenly dropping to 2 to 4 mm between 01:00 and 05:00 UTC on August 24. Later, uninterrupted extreme rainstorms

5   hit the north and coasts of the Artvin district. According to the hourly rainfall observations, the highest precipitation amounts were recorded at Hopa station during the eight hours of the flash flood day (Fig. 4b). Daily precipitation record was observed with 144.3 mm in six hours (starting at 05:00 UTC and ending at 11:00 UTC) in Hopa, and maximum hourly rainfall measured 51.5 mm at 09:00 UTC.  In Arhavi, daily total precipitation was 135.5

10  mm and reached a maximum value at 00:00 UTC with 32.4 mm. In Borçka, while daily precipitation amounts were higher (148.9 mm) than at Hopa and Arhavi, peak values of hourly precipitation intensities were lower. According to the results from these three stations, hourly precipitation reached a maximum value at 09:00 UTC in the low altitudes of the region; this implies that the precipitation was much lower in the upper sectors of the

15  mountainous area.

### 3.3 Synoptic overview

This section treats the atmospheric circulation and associated physical mechanisms that were responsible for the flash flood in the region. In order to better evaluate the phenomenology of the event, pre-existing synoptic conditions starting from August 23 were

20  investigated. At 00:00 UTC on August 23, the summer Asian monsoon low extends to the eastern Black Sea (Fig. 5a). During the summer months, in consequence of the excessive surface heating over the arid regions of the Middle East, the monsoon low expands





westward and generates the Persian trough (Alpert et al. 2004; Saaroni et al. 2010), which extends to Turkey, forming a thermal low over the eastern Mediterranean (Tyrlis et al. 2015). Besides the surface synoptic conditions, low-level moisture convergence, specific humidity content and geopotential height values of 850 hPa were extracted. It is known that

low-level moisture convergence is a good indicator for large-scale precipitation (e.g. Fragoso et al. 2012), and east of Turkey (Azerbaijan) has good synoptic precipitation conditions. In the upper levels, the presence of a weak ridge over northern Africa and through the axis over the Aegean Sea (because of the upper-level cold low over central Europe) concludes with southwesterly winds over the Artvin district (Fig. 5c).

On August 24 at 00:00 UTC, a high pressure center (HPC) over northern Russia moved to the south, located around 30° E, 60° N. While the cyclone remained almost stationary, a new cut-off cyclone occurred over the EBS (Fig. 5d). Thus, high northeasterly winds brought moisture from the Black Sea to the eastern coasts of Turkey (Fig. 5e). As a result, deep precipitation areas were observed over these regions according to the low-level

moisture convergence results. In the upper level chart (500 hPa), shifting cold core of upper level high to the south cause moving of mid-latitude low to the west, and, thus; south-westerly winds turn into the westerly together with decreasing of temperature from -7.5 °C to -10 °C (Fig. 5f).

At the start of the rainstorm (August 24, 06:00 UTC), similar surface and upper-level large-

scale circulations appeared compared with the midnight synoptic conditions (Figs 6a and 6c). Strong moisture convergence zones were detected over the flash-flood region (Fig. 6b). For this reason, thermodynamic analysis was needed to better understand the evaluation of





physical mechanisms that develop severe precipitation. Hence, as a consequence of analyzing the nearest radiosonde measurements from Samsun station (41.34 °N, 36.25 °E), instability indices such as CAPE (Convective Available Potential Energy) and LI (Lifted Index) showed that there was no strong convective activity during and before the rainstorm 5 (not shown). In order to follow the distribution convective cells and cloud droplets in a large area, it was necessary to use satellite and radar image data.

### 3.4 Satellite and radar images

Repeated temporal resolution is an excellent tool for understanding the spatial distribution of the convective cells. Therefore, SEVIRE (Spinning Enhanced Visible and Infrared 10 Imager) MSG (Meteosat Second Generation) outputs were used to examine August 24 at 06:00 UTC. It is known that 'convective storms RGB' product visualizes the particle size features of high-level cloud tops with good contrast (Kerkmann et al., 2006). Whereas yellowish cloud tops indicate opaque ice clouds with small particles, high-level opaque ice clouds with large particles are shown as reddish. The RGB product in Fig. 7a was produced 15 by assigning the brightness temperature difference (BTD) 6.2-7.3 values as the red component, the BTD 3.9-10.8 as the green component, 1.6-0.6 as the blue component. In Fig. 7a, numerous convective storms with large ice particles are shown over the EBS. On the other hand, over the land areas (e.g. Georgia) more yellowish particles are observed and this implies the storm's intensity. Separately, SYNOP observations indicate that southerly 20 winds over the coast of the EBS stations met with humid northerly flows throughout the seaside area. If the land (21 °C) and sea surface temperatures (SSTs) were sufficiently different, the convective instability and storm severity could have increased with time. As



seen in Fig. 7b, high PPI (Plan Position Indicator) reflectivity values from the radar image showed that two cores of the extreme precipitation were over the Hopa and Çayeli sub-basins.

### 3.5 Sea surface temperature (SST) analysis over Black Sea

5 The influence of SSTs on precipitation over Turkey was detail investigated by Bozkurt and Sen (2011). They found that increased SSTs led to increased precipitation of the peninsula especially downwind of the sea. Later, Kömüşçü and Çelik (2013) explained that warm Aegean SST is one of the significant causes of the development of rainstorms. In this study, exploring the role of Black Sea surface temperatures on storm development, long-term 10 (1982-2015) means of August SSTs were extracted for BS using NOAA High Resolution SST data (provided by NOAA/OAR/ESRL PSD, Reynolds et al. 2007). As seen in Fig. 8a, cold SSTs of the BS were above the latitude of 44 °N. The warmest pool of the BS in the eastern BS and SSTs exceed 27 °C in this month. During the day of the extreme event, spatial distribution of the SSTs indicates negative anomaly values in the upper 44 °N 15 latitudes (Fig. 8b). The EBS region has the highest SST anomalies and 1.5 °C higher SST variations compared with the August means for the EBS.

### 3.6 Forecasting tools: Numerical Weather Prediction (NWP) models

According to the ECMWF daily precipitation product, spatial coverage of the maximum daily precipitation values (over 160 mm) is shown in the northern Rize and northwestern 20 Artvin cities (Fig. 9a). Compared with the model output (Fig. 4a), station observations are clearly underestimated in northern Rize. On the other hand, model predictions for the

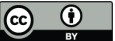

Arhavi and Borçka settlements, except Hopa, were good. With regard to Alaro's model results, the highest daily precipitation totals were well predicted only for Hopa district at 150 mm (Fig. 9b). Although precipitation forecasts of this limited-area model described Hopa well, the other two flood regions were not well predicted. Optimum spatial coverage of the daily precipitation forecasts is shown in the mesoscale WRF outputs (Fig. 9c). The problem with this model is the underestimated forecasts compared with the observation data. In TSMS, meteorologists merge the outputs of these models (the so-called "poor man ensemble") with their own experience and provide quantitative precipitation forecasts for the alert sub-regions in predefined time windows. As a consequence of this subjective prediction, TSMS and its regional weather forecast offices give alert messages related to natural hazards including severe precipitation events. These organizations also carry the responsibility for nowcasting and monitoring rainfall events. According to the main alert on August 23, 2015 at 09:00 UTC prepared by TSMS Weather Forecast Centre, very intense precipitation between 51 and 100 mm was predicted at the Rize, Artvin and Trabzon districts within 12 hours of August 24. The authorities and the public were alerted to the risk of flash flood, lightning, and landslide events.

## 4 Conclusions

The flooding event on August 24, 2015 that hit the Artvin area has been analyzed from a meteorological perspective. A large amount of precipitation fell in an area of a few square kilometers with high intensity in about 6 to 7 h, and NWP models cannot well predict such extreme events. Although alert messages were prepared by TSMS on August 23 at 09:00



UTC, 11 people died and infrastructures, buildings, private property and public goods were damaged as a result of the flash flood.

According to the synoptic conditions, when the summer monsoon frontal system extended to eastern Anatolia, its activity was enhanced. On the other hand, because of the depressive

effect of the Siberian high from the north, a cut-off low occurred over the eastern Black Sea. As a result, a slow-moving quasi-stationary cut-off low over the Black Sea increased the SSTs and more moisture was transported to the atmosphere. Thus, strong moisture convergence at low-levels (850 hPa) was observed over Artvin city. Moreover, warm humid northerly airs from the Black Sea and relatively cool southerly flows (21 °C) over

the land areas increased the instability conditions and redevelopment of the convective cells over the same region enhanced the rainfall intensity.

This paper investigated the meteorological role in an extraordinary rain event over Artvin. The synoptic and atmospheric descriptions give better knowledge of the mesoscale convective systems and the mechanisms driving torrential rains in the EBS. It is hoped that

more detailed studies will be performed on synoptic development leading to extreme summer precipitation events in EBS.

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





**Table 1 :** Description of 49 meteorological stations in the study. Stations marked by stars were used for the climatological approach.

| Station No. | Station Code | Station name | Longitude (E) | Latitude (N) | Altitude (m) | 23 Aug 2015 precip. (00-00 UTC) | 24 Aug 2015 precip. (00-00 UTC) |
|---|---|---|---|---|---|---|---|
| 1 | 17042 | Hopa* | 41.4330 | 41.4065 | 33 | 55.3 | 144.3 |
| 2 | 17045 | Artvin* | 41.8187 | 41.1752 | 613 | 0 | 1.4 |
| 3 | 18216 | Yusufeli | 41.5464 | 40.8228 | 601 | 0 | 4.4 |
| 4 | 18217 | Savsat | 42.3206 | 41.2433 | 1125 | 0 | 24.4 |
| 5 | 18218 | Ardanuc | 42.0653 | 41.1267 | 577 | 0 | 11.6 |
| 6 | 18554 | Arhavi | 41.2928 | 41.3166 | 290 | 22.4 | 135.5 |
| 7 | 18555 | Borcka | 41.6281 | 41.3750 | 190 | 35.8 | 148.9 |
| 8 | 18556 | Murgul | 41.5564 | 41.2617 | 565 | 0.2 | 42.5 |
| 9 | 17089 | Bayburt* | 40.2207 | 40.2547 | 1584 | 0.4 | 0 |
| 10 | 18219 | Demirozu | 39.8858 | 40.1639 | 1757 | 0 | 0 |
| 11 | 18557 | Aydintepe | 40.1294 | 40.3817 | 1600 | 0.6 | 0 |
| 12 | 17088 | Gumushane* | 39.4653 | 40.4598 | 1216 | 0.1 | 0 |
| 13 | 17696 | Torul (Zigana kayak m) | 39.4037 | 40.6413 | 2050 | 0 | 0 |
| 14 | 18226 | Kurtun | 39.1456 | 40.6825 | 739 | 0 | 1.5 |
| 15 | 18227 | Torul | 39.2989 | 40.5686 | 1009 | 0 | 0 |
| 16 | 18228 | Kelkit | 39.4361 | 40.1506 | 1483 | 0 | 0 |
| 17 | 18564 | Kose | 39.6578 | 40.2217 | 1635 | 0.1 | 0 |
| 18 | 18565 | Siran | 39.1289 | 40.1856 | 1490 | 3.3 | 0 |
| 19 | 17040 | Rize* | 40.5013 | 41.0400 | 3 | 28.3 | 26.2 |
| 20 | 17628 | Pazar* | 40.8993 | 41.1777 | 78 | 35.8 | 49 |
| 21 | 17713 | Camlihemsin (Ayder FI) | 41.1103 | 40.9518 | 1354 | 1.6 | 18.8 |
| 22 | 17741 | Ikizdere (Sivrikaya) | 40.7106 | 40.6711 | 1926 | 0 | 7.8 |
| 23 | 17757 | Ikizdere (Derekoy) | 40.5989 | 40.7258 | 970 | 0.4 | 37.2 |
| 24 | 17761 | Kalkandere | 40.4400 | 40.9278 | 138 | 5.7 | 75.1 |
| 25 | 17765 | Camlihemsin | 40.9942 | 41.0125 | 390 | 2.8 | 32.1 |
| 26 | 17769 | Hemsin | 40.8992 | 41.0503 | 307 | 22.3 | 21.9 |
| 27 | 17772 | Ardesen (Yesiltepe) | 41.0703 | 41.1528 | 573 | 0.4 | 0 |
| 28 | 17775 | Iyidere (Fidanlik) | 40.3319 | 40.9835 | 6 | 21.1 | 29.8 |
| 29 | 17781 | Cayeli (Teias) | 40.7417 | 41.0603 | 54 | 31.9 | 30.9 |
| 30 | 17785 | Cayeli (Kaptanpasa) | 40.7789 | 40.9583 | 483 | 15.2 | 54.1 |
| 31 | 17800 | Guneysu | 40.6083 | 40.9897 | 124 | 31.1 | 58.8 |
| 32 | 18566 | Derepazari | 40.4289 | 40.9897 | 397 | 20.1 | 38 |
| 33 | 18567 | Findikli | 41.1556 | 41.2703 | 190 | 24.7 | 62.3 |
| 34 | 18568 | Rize (Andon) | 40.5825 | 40.8711 | 615 | 12.6 | 88.8 |
| 35 | 18569 | Ikizdere (Cimil) | 40.7828 | 40.7333 | 2020 | 0.5 | 16.3 |
| 36 | 18905 | Cayeli (Bakir) | 40.7669 | 41.0408 | 100 | 32.3 | 56.5 |
| 37 | 17037 | Trabzonbolge* | 39.7649 | 40.9985 | 25 | 2.6 | 17.4 |
| 38 | 17569 | Caykara (Uzungol) | 40.4435 | 40.6193 | 1114 | 1.6 | 11.6 |
| 39 | 17626 | Akcaabat* | 39.5615 | 41.0325 | 3 | 1 | 36.6 |
| 40 | 17714 | Macka (Altindere sume.) | 39.6532 | 40.6985 | 1030 | 0.4 | 1.6 |
| 41 | 18229 | Duzkoy | 40.1339 | 40.7708 | 622 | 0.7 | 8.2 |
| 42 | 18230 | Tonya (Kalincam) | 39.2617 | 40.7803 | 1100 | 0 | 7.1 |
| 43 | 18231 | Besikduzu | 39.2144 | 41.0328 | 374 | 12 | 30.1 |
| 44 | 18232 | Hayrat (Pazaronu) | 40.4961 | 40.8858 | 367 | 17.6 | 43 |
| 45 | 18233 | Arsin | 39.9497 | 40.9486 | 169 | 0 | 14.5 |
| 46 | 18570 | Dernekpazari | 40.2719 | 40.7997 | 721 | 7 | 9.7 |


| 47 | 18571 | Koprubasi (Beskoy) | 40.1339 | 40.7710 | 975 | 14 | 17.3 |
| 48 | 18573 | Carsibasi (Yoroz) | 39.4208 | 41.0950 | 370 | 1.2 | 47.8 |
| 49 | 18574 | Surmene (Denizbilimleri) | 40.2097 | 40.9231 | 5 | 49.5 | 33.8 |

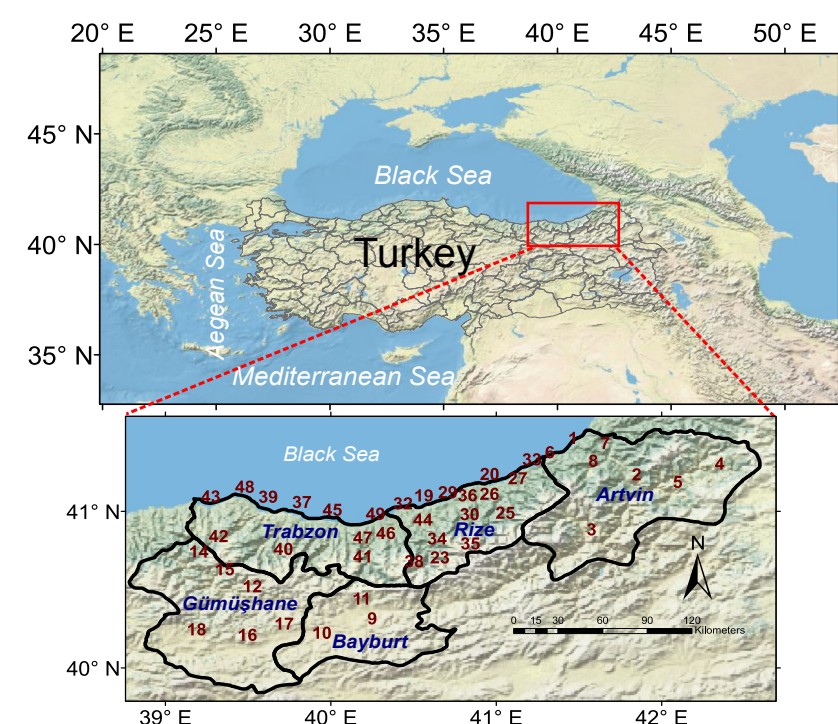

**Figure 1**. The eastern Black Sea Region included city names and borders and 49 automated

meteorological stations (Descriptions of the station numbers are explained in Table 1). The

outset shows location of the region in Turkey.



(a) (b)

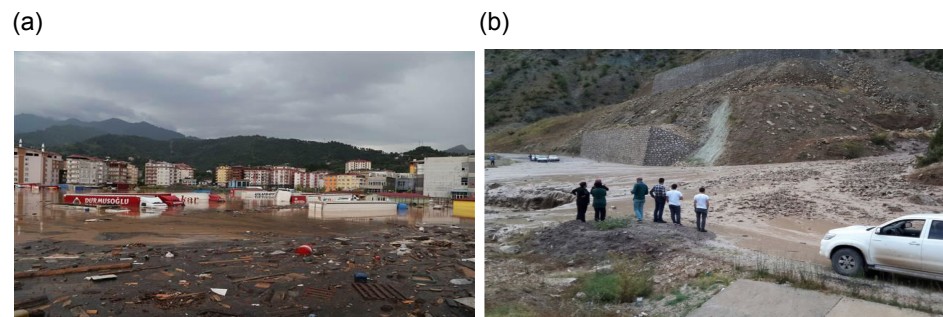

**Figure 2**. Photos showing the destructive effects of the 24 August 2015 flash-floods and landslides in: **(a)** Hopa centre flash-flood and **(b)** landslide in Hopa

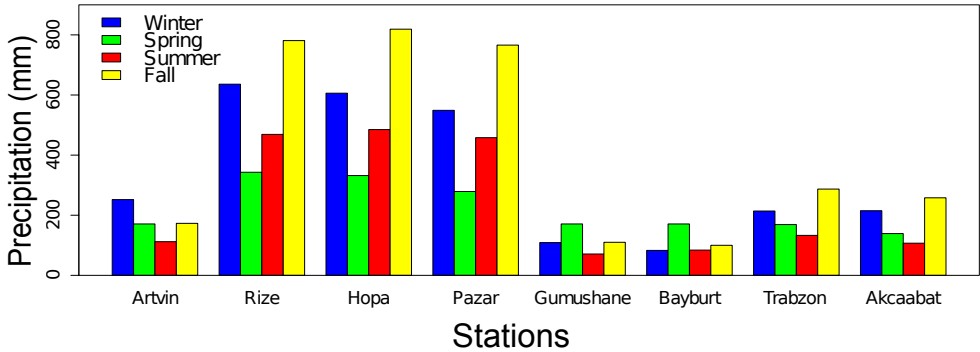

5  **Figure 3**. Long-term (1960-2014) mean of the seasonal precipitation amounts related to the eight meteorological stations in the EBS.



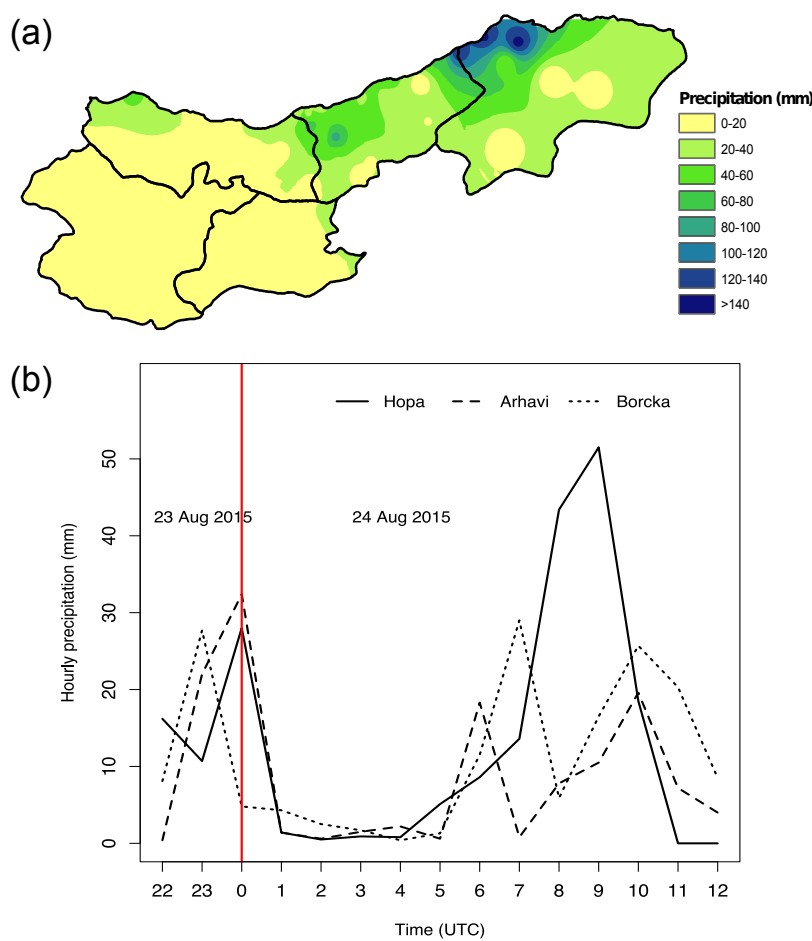

**Figure 4**. **(a)** Total daily precipitation in eastern Black Sea (00:00-24:00 UTC), 24 August

2015. The map is based on data from the same meteorological stations represented in Fig. 1

(station names are listed in Table 1). **(b)** Hourly evolution of the 24 August 2015 rainstorm

5    in Artvin, in three selected stations representing flash-flood regions



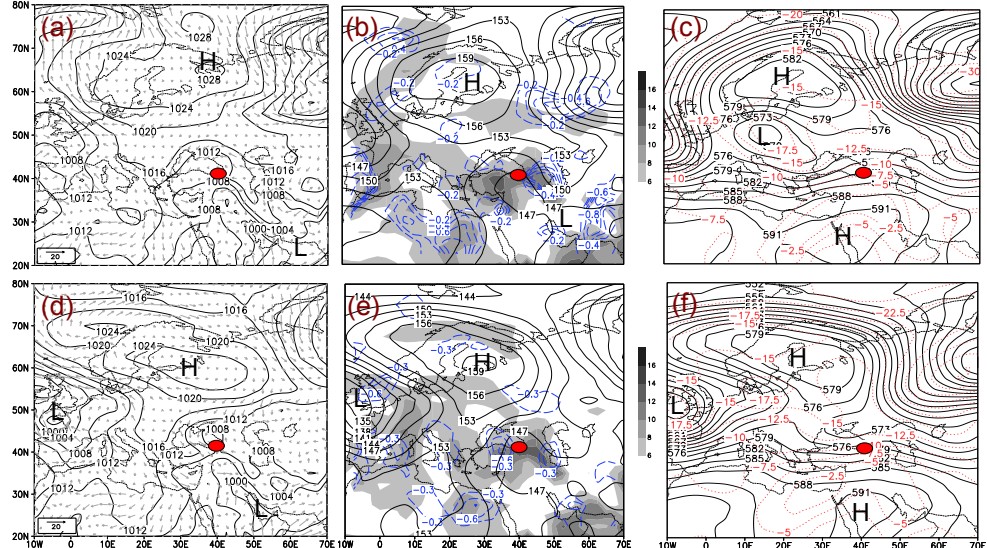

**Figure 5**. **(a)** Sea level pressure chart (lines, units in hPa) and surface winds (arrows, units in m s$^{-1}$). **(b)** Geopotential height field (units in dm), specific humidity contents (shaded in colors, units in $10^{-3}$ kg kg$^{-1}$), and moisture convergence values (dashed lines, removed positive values) of the 850-hPa level. **(c)** Geopotential height field (units in dm), and temperature values (dashed red lines, °C) of the 500-hPa level. Synoptic charts are belonging to the 23 August 2015, 00:00 UTC. The data of surface, lower and upper levels are derived from NCEP/NCAR Reanalysis. Red dot marks the studied region. **(d)** same as **(a)**, **(e)** same as **(b)**, **(f)** same as **(c)**, but for 24 August 2015, 00:00 UTC.

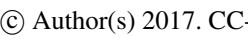


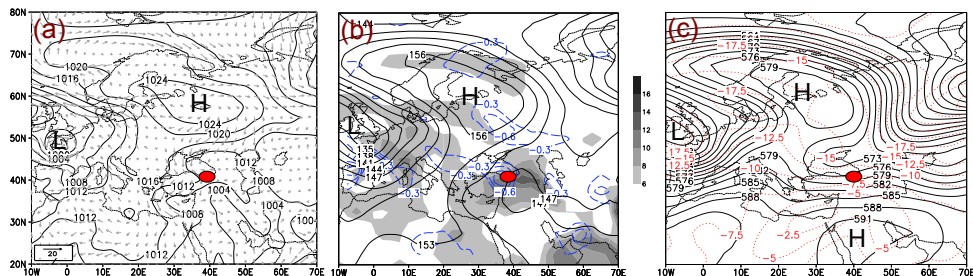

**Figure 6**. **(a)** same as Fig. 5(a), **(b)** same as Fig. 5(b), **(c)** same as Fig. 5(c), but for 24 August 2015, 06:00 UTC.

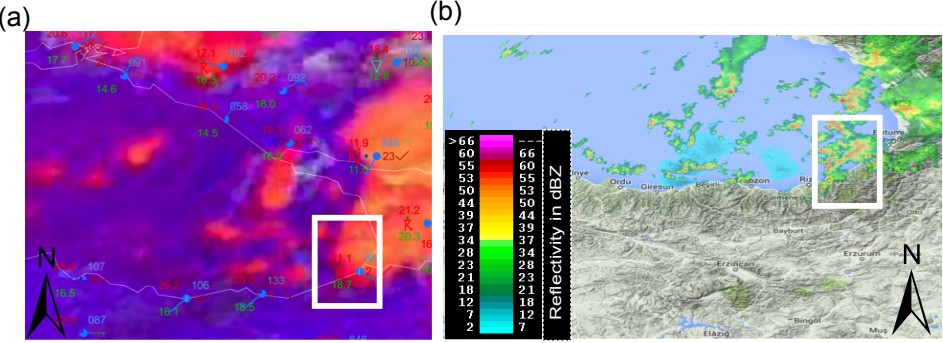

**Figure 7**. Satellite and radar images on 24 August 2015, 06:00 UTC. **(a)** Convective storm RGB product from SEVIRE MSG (Meteosat Second Generation) together with SYNOP observations. **(b)** Radar PPI (Plan Position Indicator) image of the EBS region. Sources: (a) EUMETRAIN (http://www.eumetrain.org/) (b) Turkish State Meteorological Service (www.mgm.gov.tr)

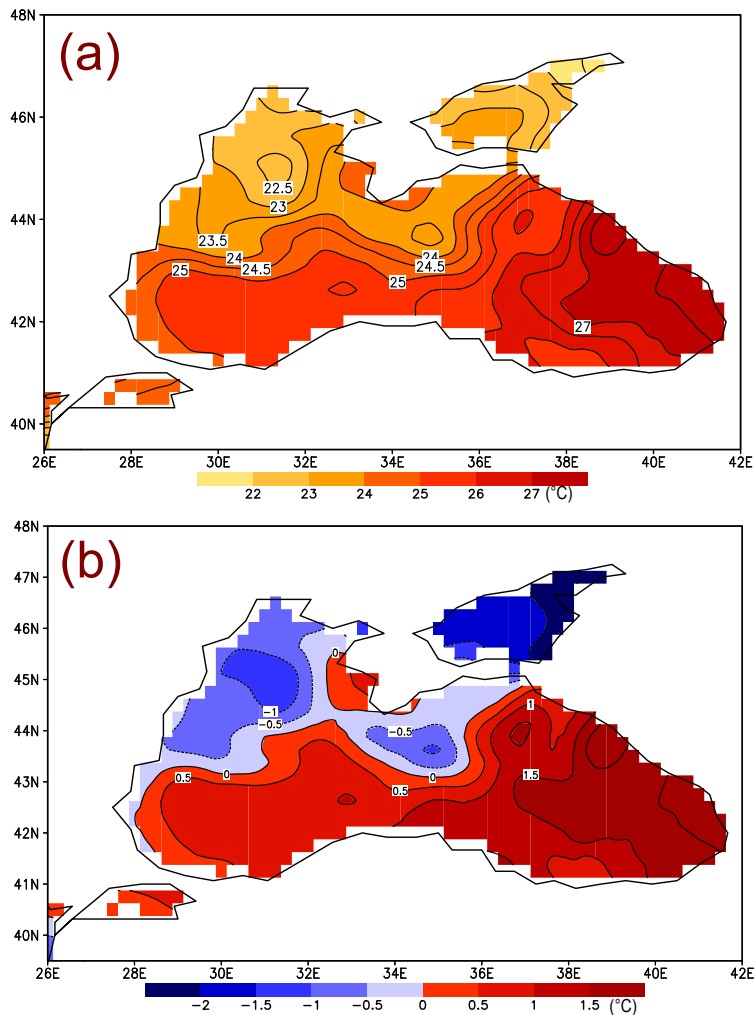

**Figure 8**. **(a)** Spatial distribution of the long-term (1960-2014) mean of august sea surface

temperatures (SSTs) over Black Sea. **(b)** Anomaly values of the 24 august daily mean SSTs

when compared with long-term august mean SSTs. The SST Reanalysis data is derived

from NOAA High Resolution SST (from their website is http://www.esrl.noaa.gov/psd).


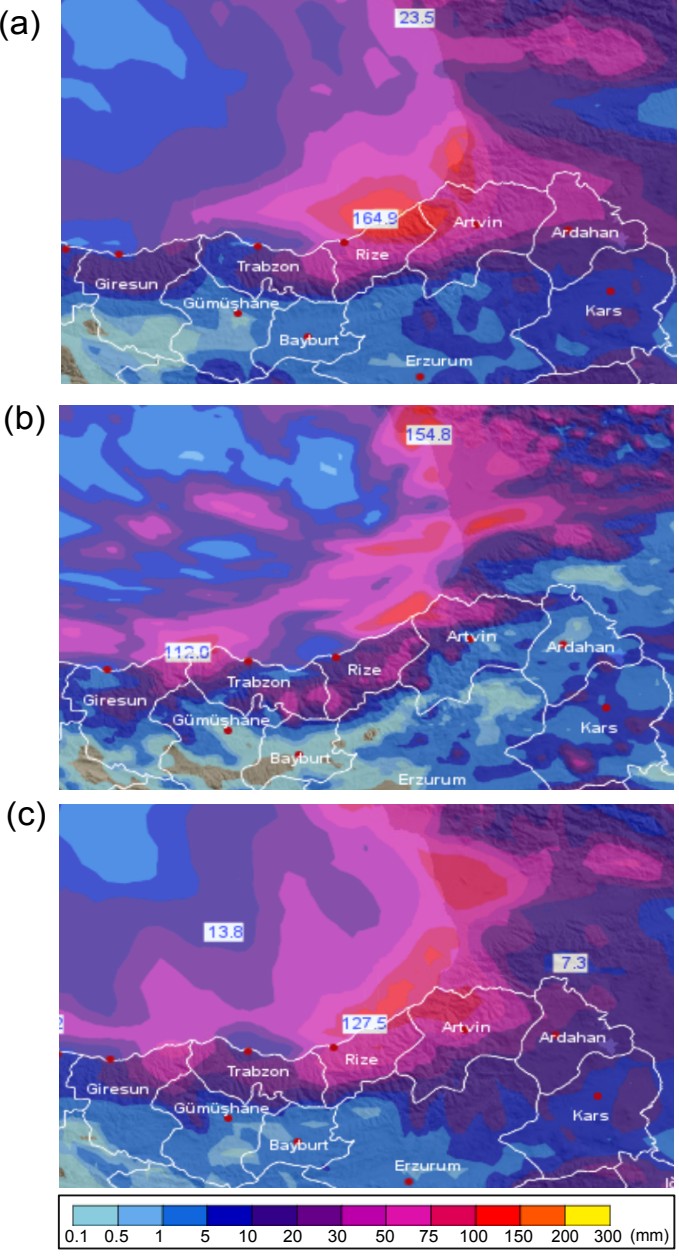

**Figure 9**. Numerical Weather Prediction (NWP) precipitation forecasts for the 24-h daily

precipitation totals belonging to the 24 August 2015 in the EBS region. **(a)** for ECMWF **(b)**

for ALARO and **(c)** for WRF. Sources: (a-c) Turkish State Meteorological Service (TSMS)