# Peer review of "Meteorological analysis of flash floods in Artvin (NE Turkey) on"

_Natural Hazards and Earth System Sciences, 2016_

## Referee Comment (RC1) · Anonymous Referee #1 · 24 Mar 2017

**General comments**

The paper deals with a meteorological analysis of flash flood in Artvin (NE Turkey) on 24 August 2015. Apart some technical errors below reported, the paper is understandable and the author describes the topic with a good introduction well referenced. He also analysed the meteorological episode using radar, satellite images, weather models and synoptic meteorological stations, giving an overall comprehension of the meteorological phenomenon occurred in the NE of Turkey that caused a flash flood with a landslide. Hence, the paper is reasonably addressed in the aims of NHESS.

**Specific comments**

P2, L16-19: While I understood the second type of flood, it is not clear the first type. Please, clarify it.

P6, L3: I think the comparison is between Figure 4a and 9: can you show them with the same colour legend in the investigated area? The differences between models and observations would be better appreciated.

In fact, in P12-L4, when you say: "Optimum spatial coverage…" It's difficult to spot with different maps and colours.

P5, Section 2.3: Please, add some references about the NWP models, in particular some physics features about the WRF (used version) and the ALARO models.

P6, L13: What do you mean with "moderate climates"?

P6, L17: you wrote: "the highest amount of precipitation is observed during wet and dry seasons". It confuses me: how could it be in dry season, if we are talking of the highest amount?

Conclusions: It is well known that these extreme events are usually studied from a hydro-meteorological point of view and, as future developments, it would be interesting to see hydro-meteorological models coupled together in order to forecast the exceeding a warning threshold at least, since the meteorological warning has been correctly issued by the TSMS.

**Technical corrections:**

P1, L22: I suggest: "…flood damages in the Artvin area."

P2, L5-10: I suggest: "For example, just one flash flood in 2002…"

P2, L9: I suggest: "For example, just a single flash flood caused €1.2 billion Euro damages in the Gard region of France in 2002 (Huet et al., 2003), €300 million Euro damages in the Pinios (Greece) flash flood during 1994 (Gaume et al., 2008), €65 million Euro economic losses in the Magorala (Spain) flash flood in 2000 (Llasat et al., 2001), and €4.6 million Euro in the 2007 Mastroguglielmo (Italy) flash flood event (Aronica et al., 2008).

P3, L9: Remove "a" before "slow-moving"

P3, L13: I suggest "triggered landslides"

P3, L16: Remove "the" before "rainfall"

P4, L7: Remove the apostrophe after "dollars"

P5, L2: I suggest "meteorological stations"

P5, L4: I suggest: "and to" instead of "as well as"

P5, L9: I suggest "as the previous day atmospheric…"

P5, L19: I suggest: "In the Alaro meteorological model…"

P6, L16: I suggest: "due to…" instead of "because of…"

P7, L4: I suggest to replace "instead of" with "while"

P7, L9-10: I suggest to remove this sentence: "Nevertheless…"

P7, L16: Replace "those" with "the"

P8, L2: Replace: to with "from"

P8, L3: Do you mean "dropping from 4 to 2"?

P8, L7: I suggest: "The maximum daily precipitation value was…"

P9, L6: I suggest: "eastern" instead of "east to"

P9, L9: "and through the axis": this sentence is not clear, please clarify it.

P9, L16: add "the" before "moving". Remove semicolon after "thus"

P10, L5: "(not shown)": do you mean not shown in the text?

P10, L19: I suggest "the storm intensity"

P12, L1: I suggest: to the Alaro model"

P12, L10: I suggest "issue" instead of "give"

P12, L20: I suggest "6 to 7 hours"

P13: L12: I suggest to move this sentence at the beginning of the conclusion section.

P20, L6: I suggest: "in °C)"

---

## Referee Comment (RC2) · Anonymous Referee #2 · 11 May 2017

the study presented the unique topic in this location. while in this area has lots of flood or lanslide events, there is no investigation have been made in this area using meteorologic aspect. also the author investigated impact of sea water temperature, low moisture convergance objective on the flood and lanslides. and precipitaion values and model which used in this study were compared each other. this is important for the estimation. besides high density meteoriologic network has increased last years in this area, and it enables spatial and temporally distribution.

under these conditions the study has topical, but some polish on grammer and English can be done. also floatation and rain background in this area can be presented with compare other national and international datas.

[Figure]

2017.

---

## Referee Comment (RC3) · Anonymous Referee #3 · 20 May 2017

This paper investigates the meteorologic conditions before the flash flood event occurred in the city of Artvin on 24th of Aug 2015 in north-eastern part of Turkey. This region is prone to flash flood events which lead to landslides in the region due to intense storms and steep topography. Hence, the outcome of this study is important for decision makers to establish an early-warning system.

The paper is suitable for Natural Hazards and Earth System Sciences. I have minor points which should be addressed. I highly recommend to send it a professional editing service.

Moderate points:

- At the end of abstract, I recommend one sentence as a take-home message (general message).

- Similarly, in the Conclusion Section, please add a few sentences as a take-home message for decision makers to emphasize the applicability of the outcomes of this study.

Minor Points:

P1. L12. ...total accumulated rainfall AMOUNTS of 136, 69, and 109 mm WERE measured....

P1. L22. Delete 'the' before Artvin.

P2. L2. Insert 'the' before warm. ......on THE warm Mediterranean Sea.....

P2. L6. Delete 'the' before 2002.

P2. L6. Delete Euro sign (€. ...caused 1.2 billion damages...... Similar, also correct: P2.L7, P2.L8, P.2L9....

P2. L9. Delete 'the' before 2000. ....flood in 2000....

P2. L10. WORD CHOICE. My recommendation: .....it is necessary to IMPROVE OUR CURRENT UNDERSTANDING about the.......

P2. 16. Re-write. My recommendation: Depending on the catchment characteristics, mainly two types of flood occur in Turkey.

P2. L18. COMMA. Insert a comma after 'affected'.

P2. L18. CAPITALIZATION. Capitalize 'river'. ....of the Meric River...

P3. L5. Re-write. My recommendation: ...in Antalya, a coastal city located on the Mediterranean Sea.

P3. L8. Re-write. My recommendation: .....investigated the hydrometerological role of floods occurred during 7-10 September, 2010 in the Marmara Region.

P3. L15-17. Re-write. My recommendation: The underlying geology of the EBS is generally consists of semi-permeable volcanic rocks which reduce infiltration and enhance runoff production (XXXX).

P3. L17-19. Re-write. My recommendation: The north-eastern coastal parts of Turkey, regions located on the windward slopes of the EBS facing the Black Sea, receives more than 2000 mm of annual precipitation which is the wettest part of the country.

P3. L19-21. Re-write. My recommendation: The large mountainous area which extends through the Black Sea, and slope instability due to steep gradients as well as intense rainfall result in flash floods and landslides and threaten the settlements in the EBS region.

P4. L1. VERB. . . .facilitate. . .

P4. L7. TYPO at the end of dollars. . . .dollars' Delete the apostrophe.

P4. L8. WORD CHOICE. . . ..the DETRIMENTAL EFFECTS of floods for. . ...

P4. L11. WORD CHOICE. . . .the aim of this research is TO FOCUS on. . ...

P4. L20. Insert a comma after synoptic. . . .synoptic, and . . . . . .

P4. L22. WORD CHOICE. . . ..with WEATHER forecasts. . ..

P5. L4. Insert 'to' before retrieve. . . ..as well as TO retrieve. . . . . .

P5. L16-19. Re-write.

P6. L2. Insert a comma after 'domain'. . . .domain, and . . ...

P6. L9. Re-write. My recommendation: This mountain chain extends parallel to the Black Sea and . . ...

P6. L11-12. Re-write. My recommendation: . . .the region also experiences orographic effect on precipitation.

P6. L14-15. Re-write. My recommendation: The rain shadow effect on the lee side of

the mountainous area CAUSES a more. . .

P7. L2. WORD CHOICE. . . ..(MAP) VARIES from. . . . . .

P7. L6. Explain MCS. Describe acronym 'MCS'.

P7. L7. WORD CHOICE. . . .were OBSERVED AT Hopa, Rize, and Pazar with . . ...

P7. L13. Insert a comma after 'Hopa'. . . ..Hopa, and . . ...

P7. L17. Insert a comma after 'Arhavi'. . . ..Arhavi, and . . ...

P7. L20. WORD CHOICE. . . .Another coastal station, Arhavi. . . ..

P7. L22. Re-write. Describe it. Temporal distribution of WHAT? Temporal distribution of XXXXXXX that. . . . . .

P8. L1. Insert 'THE'before midday. . . . . . .at THE midday on the. . ..

P8. L2. REPLACE. Hourly observations AT the three stations. . ..

P8. L2. REPLACE. . . ..increased FROM 27 to 32. . ..

P8. L3. REPLACE. . . ..DROPPED to 2-4 mm. . ..

P8. L6. DELETE 'station'. . . ..at Hopa during. . . . . .

P8. L19 CHECK. I am not sure 'phenomenology' is the correct word there?

P9. L4-7. Re-write. (Azarbijan). Make sure that a reader should understand that Azerbaijan is another country that locates east of Turkey.

P9. L17-18. Re-write. My recommendation: . . ..with a decrease in temperature from . . ...

P10. L1. VERB. . . ..that developed severe. . . . . .

P10. L4. REPLACE. . . ..activity before and during . . . . ...

[Figure]

P10. L11. ...were used to examine THE ATMOSPHERIC CONDITIONS ON 24 August......

P10. L17-19. Re-write. You do not need to say more yellowish. On the other hand, more intense storms were observed over the land areas such as Georgia (Fig.7a).

P11. L5. WORD CHOICE. ....was investigated IN DETAIL by.....

P11. L9. WORD CHOICE. ...the role of SSTs of the Black Sea on........

P11. L10. INSERT 'the'. ....for THE BS....

P11. L12. WORD CHOICE. ....were NORTH OF the latitude of 44°N.

P11. L13. VERB TENSE. Use PAST TENSE. ....exceedED ......

P11. L14. WORD CHOICE. ..... values in NORTH OF 44°N latitude.

P11. L20. VERB TENSE. ........station observations WERE clearly.........

P12. L1. Describe Alaro model.

P12. L5-7. Re-write the sentence.

P12. L12. VERB TENSE. ...offices GAVE alert messages.......

P12. 14. Insert a comma after 'Artvin'. ....Artvin, and Trabzon....

P13. L7. WORD CHOICE. ...was transported FROM THE SEA to the atmosphere.

Figure 1. Narrower region for Turkey map. Show Georgia and Azerbaijan as countries.

Figure 2. In the caption: Hopa CITY centre......

Figure 4. In the caption: ...in THE eastern Black Sea.......

Figure 5. In the caption: I recommend using following: ....units in g kg-1).......

Figure 5. L7. Insert a space after 2015, 00:00 UTC......

Figure 8. L2. Mean of August. . ..

L3. . . ...over THE Black. . ... 24 August. . ..

L4. . . .long-term August . . .. . . ...data ARE derived. . ...

Figure 9. Delete comma after region. . . ..region (a) for. . ...

---

## Author Response (AR1)

**Answer to anonymour Referee #1**

I thank the reviewer for his/her constructive comments. According to your reviews, I rearranged the manuscript and answered your questions as follows.

**Reviewer quote 1:** While I understood the second type of flood, it is not clear the first type. Please, clarify it (P2, L16-19)

**Answer 1:** To explain the first type of flood, I added a sentence "In the first type, river basins respond rapidly to intense rainfall because of steep slopes, impermeable surfaces, saturated soils, or because of anthropogenic forcing to the natural drainage" to the second paragraph of the introduction section.

**Reviewer quote 2:** I think the comparison is between Figure 4a and 9: can you show them with the same colour legend in the investigated area? The differences between models and observations would be better appreciated (P6,L3).

**Answer 2:** Thanks for your comments. In accordance with your explanations, I rearranged Fig. 4a colour legend similar that of Fig. 9.

**Reviewer quote 3:** When you say: "Optimum spatial coverage…" It's difficult to spot with different maps and colours. (P12-L4).

**Answer 3:** As you mentioned above, it can be difficult to distinguish the optimum spatial coverage. However, when we focused on the spatial distribution of the measured precipitation (Fig. 4a) data, we can see the highest daily precipitation totals are only shown in the seaside stations of Artvin and the other two NWP model forecasts do not well coincide with observation values.

**Reviewer quote 4:** Please, add some references about the NWP models, in particular some physics features about the WRF (used version) and the ALARO models (P5, Section 2.3)

**Answer 4:** To give more details of the physics of the NMM-WRF, I added some sentences at the end of the first paragraph of the Section 2.3 as follows:

"The mesoscale NWP system of Non-hydrostatic Mesoscale Model (NMM) core of the Weather Research and Forecasting (WRF) is developed by the National Oceanic and Atmospheric Administration (NOAA)/National Centers for Environment Prediction (NCEP), NMM-WRF is a fully compressible, non-hydrostatic mesoscale model with a hydrostatic option (Janjic, 2003). The model uses a terrain-following hybrid sigma-pressure vertical coordinate. The grid staggering is the Arakawa E-grid. The model uses a forward-backward scheme for horizontally-propagating fast waves, an implicit scheme for vertically-propagating sound waves, the Adams-Bashforth scheme for horizontal advection, and the Crank-Nicholson scheme for vertical advection. The dynamics conserve a number of first and second order quantities including energy and enstrophy."

For the Alaro model, a brief explanation together with its references was added to the Section 2.3 as follows:

"For the regional weather forecasts, the Alaro meteorological model has been designed to be run at convection-permitting resolutions. The key concept is in the precipitation and cloud scheme called Modular Multiscale Microphysics and Transport (3MT) developed by Gerard and Geleyn (2005), Gerard (2007), and Gerard et al. 2009. In the usage of the Alaro by TSMS, whereas the outer domain has grid spacing of 10 km, the inner domain has almost 5 km of grid spacing as well as 60 vertical levels."

**Reviewer quote 5:** What do you mean with "moderate climates"? (P6, L13)

**Answer 5:** "Moderate climates" were changed to "more dry climates" term in the text.

**Reviewer quote 6:** you wrote:"the highest amount of precipitation is observed during wet and dry seasons". It confuses me: how could it be in dry season, if we are talking of the highest amount?

**Answer 6:** "… In terms of precipitation values,…" sentence was rephrased and instead, "When compared with the other regions, highest winter and summer precipitation totals are observed in this part of Turkey due to the interactions of synoptic weather patterns and orographic lifting." term was added.

**Technical corrections:**

P1, L22: I suggest: "…flood damages in the Artvin area."  It was corrected.
P2, L5-10: I suggest: "For example, just one flash flood in 2002…" It was corrected.
P2, L9: I suggest: "For example, just a single flash flood caused €1.2 billion Euro damages in the Gard region of France in 2002 (Huet et al., 2003), €300 million Euro damages in the Pinios (Greece) flash flood during 1994 (Gaume et al., 2008), €65 million Euro economic losses in the Magorala (Spain) flash flood in 2000 (Llasat et al., 2001), and €4.6 million Euro in the 2007 Mastroguglielmo (Italy) flash flood event (Aronica et al., 2008). It was corrected.
P3, L9: Remove "a" before "slow-moving" It was corrected.
P3, L13: I suggest "triggered landslides" It was corrected.
P3, L16: Remove "the" before "rainfall" It was corrected.
P4, L7: Remove the apostrophe after "dollars" It was corrected.
P5, L2: I suggest "meteorological stations" It was corrected.
P5, L4: I suggest: "and to" instead of "as well as" It was corrected.
P5, L9: I suggest "as the previous day atmospheric…" It was corrected.
P5, L19: I suggest: "In the Alaro meteorological model…" It was corrected.
P6, L16: I suggest: "due to…" instead of  "because of…" It was corrected.
P7, L4: I suggest to replace "instead of" with "while" The sentence was rearranged.
P7, L9-10: I suggest to remove this sentence: "Nevertheless…" It was corrected.
P7, L16: Replace "those" with "the" It was corrected.
P8, L2: Replace: to with "from" It was corrected.
P8, L3: Do you mean "dropping from 4 to 2"? It was corrected.
P8, L7: I suggest: "The maximum daily precipitation value was…" It was corrected.
P9, L6: I suggest: "eastern" instead of "east to" It was corrected.
P9, L9: "and through the axis": this sentence is not clear, please clarify it. The sentence was rearranged.
P9, L16: add "the before "moving". Remove semicolon after "thus" It was corrected.
P10, L5: "(not shown)": do you mean not shown in the text? Temp diagram was not shown

P10, L19: I suggest "the storm intensity" It was corrected.
P12, L1: I suggest: to the Alaro model" It was corrected.
P12, L10: I suggest "issue" instead of "give" It was corrected.
P12, L20: I suggest "6 to 7 hours" It was corrected.
P13, L12: I suggest to move this sentence at the beginning of the conclusion section. Done
P20, L6: I suggest: "in °C)" It was corrected.

**Answer to anonymour Referee #2**

Thanks a lot for your valuable comments for the manuscript. According to your comments related to precipitation climate of region and comparison with the other region, I gave detail information for the seasonal precipitation distribution and synoptic and orographic mechanisms that cause precipitation, and I gave a reference to emphasis the awareness of the regional precipitation than the other regions precipitation distribution in Section. 3.1.
Also, in the direction of your comments, my manuscript was sent to a native English speaker and some grammatical errors were rearranged.

**Answer to anonymour Referee #3**

I thank the reviewer for his/her constructive comments. According to your reviews, I rearranged the moderate and minor points as follows:
Moderate Points:

**Reviewer quote 1:** At the end of abstract, I recommend one sentence as a take-home message (general message)

**Answer 1:** According to your comments, i added "This study supports conventional weather analysis, satellite images, and forecast model output to alert forecasters to the potential for heavy rainfall." Sentence at the end of the abstract.

**Reviewer quote 2:** Similarly, in the Conclusion Section, please add a few sentences as a take-home message for decision makers to emphasize the applicability of the outcomes of this study

**Answer 2:** At the end of the Conclusion Section, "The synoptic and atmospheric descriptions give better knowledge of the mesoscale convective systems and the mechanisms driving torrential rains in the EBS. It is hoped that more detailed studies will be performed on synoptic development leading to extreme summer precipitation events in EBS." Sentence was corrected and added.

**Minor Points:**

P1. L12. ...total accumulated rainfall AMOUNTS of 136, 69, and 109 mm WERE measured. . .. It was corrected.
P1. L22. Delete 'the' before Artvin. Done.
P2. L2. Insert 'the' before warm. . . .. . .on THE warm Mediterranean Sea. . . ... Done.
P2. L6. Delete 'the' before 2002. Done.
P2. L6. Delete Euro sign ( . . . .caused 1.2 billion damages. . .. . . Similar, also correct: P2.L7, P2.L8, P.2L9. . .. Done
P2. L9. Delete 'the' before 2000. . . ..flood in 2000. . .. Done.
P2. L10. WORD CHOICE. My recommendation: . . ...it is necessary to IMPROVE OUR CURRENT UNDERSTANDING about the. . .. . . It was corrected.
P2. 16. Re-write. My recommendation: Depending on the catchment characteristics, mainly two types of flood occur in Turkey. It was corrected.
P2. L18. COMMA. Insert a comma after 'affected'. Done
P2. L18. CAPITALIZATION. Capitalize 'river'. . . ..of the Meric River. . . Done.
P3. L5. Re-write. My recommendation: . . .in Antalya, a coastal city located on the Mediterranean Sea. Done.
P3. L8. Re-write. My recommendation: . . ...investigated the hydrometerological role of floods occurred during 7-10 September, 2010 in the Marmara Region. Done.
P3. L15-17. Re-write. My recommendation: The underlying geology of the EBS is generally consists of semi-permeable volcanic rocks which reduce infiltration and enhance runoff production (XXXX). It was corrected.
P3. L17-19. Re-write. My recommendation: The north-eastern coastal parts of Turkey, regions located on the windward slopes of the EBS facing the Black Sea, receives more than 2000 mm of annual precipitation which is the wettest part of the country. It was corrected.
P3. L19-21. Re-write. My recommendation: The large mountainous area which extends through the Black Sea, and slope instability due to steep gradients as well as intense rainfall result in flash floods and landslides and threaten the settlements in the EBS region. It was corrected.

P4. L1. VERB. . . .facilitate. . . Done.
P4. L7. TYPO at the end of dollars. . . .dollars' Delete the apostrophe. Done.
P4. L8. WORD CHOICE. . . ..the DETRIMENTAL EFFECTS of floods for. . ... Not changed
P4. L11. WORD CHOICE. . . .the aim of this research is TO FOCUS on. . ... Not changed
P4. L20. Insert a comma after synoptic. . . .synoptic, and . . ... . Done.
P4. L22. WORD CHOICE. . . ..with WEATHER forecasts. . . . Done.
P5. L4. Insert 'to' before retrieve. . . ..as well as TO retrieve. . . .. . Done
P5. L16-19. Re-write. Section 2.3 was rewritten according to the comments of the Reviewer 1.
P6. L2. Insert a comma after 'domain'. . . ..domain, and . . ... Done
P6. L9. Re-write. My recommendation: This mountain chain extends parallel
to the Black Sea and . . ... Done
P6. L11-12. Re-write. My recommendation: . . .the region also experiences
orographic effect on precipitation. Not changed
P6. L14-15. Re-write. My recommendation: The rain shadow effect on the lee
side of the mountainous area CAUSES a more. . . Done
P7. L2. WORD CHOICE. . . ..(MAP) VARIES from. . .. . Done
P7. L6. Explain MCS. Describe acronym 'MCS'. It was explained in Introduction Section
P7. L7. WORD CHOICE. . . .were OBSERVED AT Hopa, Rize, and Pazar with . . ... Done
P7. L13. Insert a comma after 'Hopa'. . . ..Hopa, and . . ... Done
P7. L17. Insert a comma after 'Arhavi'. . . ..Arhavi, and . . ... Done
P7. L20. WORD CHOICE. . . .Another coastal station, Arhavi. . ... Done
P7. L22. Re-write. Describe it. Temporal distribution of WHAT?
Temporal distribution of XXXXXXX that. . .. . . It was corrected by adding "precipitation" after Temporal
P8. L1. Insert 'THE'before midday. . .. . .at THE midday on the. . .. Done
P8. L2. REPLACE. Hourly observations AT the three stations. . .. Done
P8. L2. REPLACE. . . ..increased FROM 27 to 32. . .. Done
P8. L3. REPLACE. . . ..DROPPED to 2-4 mm. . .. Done
P8. L6. DELETE 'station'. . . ..at Hopa during. . .. . Done
P8. L19 CHECK. I am not sure 'phenomenology' is the correct word there? Not changed
P9. L4-7. Re-write. (Azarbijan). Make sure that a reader should understand
that Azerbaijan is another country that locates east of Turkey. Not changed
P9. L17-18. Re-write. My recommendation: . . ..with a decrease in temperature
from ..... Done
P10. L1. VERB. . . ..that developed severe. . .. . Done
P10. L4. REPLACE. . . ..activity before and during . . .. . .. Done
P10. L11. . . .were used to examine THE ATMOSPHERIC CONDITIONS ON
August. . ... Done
P10. L17-19. Re-write. You do not need to say more yellowish. On the other hand,
more intense storms were observed over the land areas such as Georgia (Fig.7a). Done
P11. L5. WORD CHOICE. . . ..was investigated IN DETAIL by. . ... Done
P11. L9. WORD CHOICE. . . .the role of SSTs of the Black Sea on. . .. . . ... Not changed
P11. L10. INSERT 'the'. . . ..for THE BS. . .. Done

P11. L12. WORD CHOICE. . . ..were NORTH OF the latitude of 44$^{\circ}$N. Done
P11. L13. VERB TENSE. Use PAST TENSE. . . ..exceedED . . .. . Done

P11. L14. WORD CHOICE. . . ... values in NORTH OF 44$^{\circ}$N latitude. Done.
P11. L20. VERB TENSE. . . .. . ..station observations WERE clearly. . .. . ... Done.
P12. L1. Describe Alaro model. Detail description of the Alaro were given in Section 2
P12. L5-7. Re-write the sentence. Not changed

P12. L12. VERB TENSE. . . .offices GAVE alert messages. . .. . ..instead of GAVE, I used issued term.

P12. 14. Insert a comma after 'Artvin'. . . ..Artvin, and Trabzon. . .. Done.
P13. L7. WORD CHOICE. . . .was transported FROM THE SEA to the atmosphere. Done.
Figure 1. Narrower region for Turkey map. Show Georgia and Azerbaijan as countries. Not changed
Figure 2. In the caption: Hopa CITY centre. . ... Done.
Figure 4. In the caption: . . .in THE eastern Black Sea. . .. . . Done.
Figure 5. In the caption: I recommend using following: . . ..units in g kg-1). . ... Done
Figure 5. L7. Insert a space after 2015, 00:00 UTC. . ... Done.
Figure 8. L2. Mean of August. . .. Done.
L3. . . ...over THE Black. . ... 24 August. . .. Done.
L4. . . .long-term August . . .. . . ...data ARE derived. . ... Done.
Figure 9. Delete comma after region. . . ..region (a) for. . ... Done.

[revised manuscript text omitted]

mac baltaci 20/5/2017 01:46